# Interests of Google users in information pollution terminologies: an infodemiology study

Thiago Cruvinel[1], Olívia Santana Jorge[1], Matheus Lotto[1], Bruna Nogueira[1], Ana Maria Jucá[1] and Agnes Cruvinel[2]

[1] Department of Pediatric Dentistry, Orthodontics and Public Health, Bauru School of Dentistry, Bauru, São Paulo, Brazil
[2] Department of Medicine, Bauru School of Medicine, Bauru, São Paulo, Brazil



Corresponding author
Agnes Cruvinel,
agnescruvinel@usp.br

## ABSTRACT

**Aim:** This study aimed to identify the interests of Google users in the terms "fake news," "misinformation," "disinformation," and "conspiracy theory," particularly in specific health-related contexts.

**Methods:** This longitudinal and retrospective ecological study examined computational metadata concerning the interests of Google users from 25 countries regarding the search terms "fake news," "misinformation," "disinformation," and "conspiracy theory." Initially, relative search volume (RSV) data for these four topics were extracted from Google Trends, encompassing all categories for the period between January 2004 and March 2025. The data underwent seasonal decomposition to identify the trend, seasonal, and residual components of the collected time series, using Python 3 programming libraries within the Google Colaboratory interface. The Mann–Kendall test was subsequently applied to assess the significance of the observed trends. Additionally, search queries were qualitatively evaluated to identify health-related ones. Lastly, Spearman correlation analyses were conducted to examine the relationship between the proportion of health-related queries and both Internet penetration and mean schooling levels in the selected countries. Statistical significance was established at $P < 0.05$.

**Results:** Searches for "fake news" showed an increasing trend in all countries, while "misinformation" followed a similar pattern except in France and Japan. Interest in "disinformation" increased in most countries but decreased in Italy and showed no trend in France and the United States. For "conspiracy theory," decreasing trends were observed in eight countries and increasing trends in 16. Furthermore, a total of 52 health-related queries were identified, with seven linked to "misinformation" in 10 countries, five to "disinformation" in seven countries, 16 to "fake news" in 15 countries, and 24 to "conspiracy theory" in 17 countries, primarily related to COVID-19 ($n = 44$; 84.6%). Finally, a significant positive correlation was found only between the percentage of health-related queries and internet penetration for "misinformation" (rs = 0.48, $P = 0.017$).

**Conclusions:** Overall, these findings support the increasing interest of Google users in terms related to information pollution over time, although the association of these terms with health-related queries appears limited. This underscores the importance of implementing media education—particularly in schools—as a long-term strategy to promote a better understanding of key concepts among the population and to encourage the pursuit of information from diverse and reliable sources. Moreover, international organizations should actively monitor and regulate the emergence and

spread of new terminology related to information pollution, especially in the context of growing internet penetration worldwide.

## INTRODUCTION

The term "fake news" gained prominence during the 2016 United States presidential election, drawing significant attention to issues concerning truth, facticity, and the integrity of information in journalism and political communication (*Bovet & Makse, 2019*; *Egelhofer et al., 2020*). In the health domain, the term became increasingly relevant during the COVID-19 pandemic, which triggered a global surge in the dissemination of information—both accurate and misleading—regarding the coronavirus. Consequently, the scientific literature has sought to clarify this complex landscape through a more nuanced terminology, recognizing distinct but interrelated phenomena of information disorder (*Wardle & Derakhshan, 2017*; *Ireton & Posetti, 2018*; *Douglas et al., 2019*; *Molina et al., 2019*; *Lim, 2020*; *United Nations Development Programme, 2022*). This framework generally encompasses four primary categories:

(a) misinformation—inaccurate or misleading content shared without intent to cause harm, which may include elements of truth and is often presented as factual;

(b) fake news—fabricated or distorted content intentionally designed to mislead, frequently employing sensationalist headlines, decontextualized information, or omitted facts, to benefit the disseminator;

(c) disinformation—deliberately false information created and disseminated with the explicit intention of causing harm to individuals, groups, institutions, or nations; and

(d) conspiracy theories—explanatory narratives attributing significant social or political events to covert operations orchestrated by powerful actors, typically lacking empirical support. Notably, fake news can be conceptualized both as a specific subcategory of disinformation and as a rhetorical strategy employed to undermine the credibility of journalism and professional communication (*Egelhofer & Lecheler, 2019*). The challenge of managing the so-called infodemic becomes particularly acute in the health sector, where scientifically grounded messages are often co-opted into politicized narratives and online discourse lacking evidentiary support.

Although the circulation of terms related to information pollution in mass media and social media is well documented, a significant knowledge gap persists regarding public interest in these terms over time. The lack of empirical data on the popularization and usage of such terminology in communication studies is concerning, as their indiscriminate adoption may inadvertently reinforce the phenomena they aim to describe, along with their associated consequences, both in general contexts and in specific domains such as

health. In this scenario, the public's ability to distinguish factual information from misinformation is undermined by the erosion of boundaries between facts and beliefs—an ambiguity strategically exploited by political actors and their affiliates to discredit opposing viewpoints within an increasingly polarized digital environment, particularly in the face of content overabundance characteristic of contemporary digital ecosystems (*Allcott & Gentzkow, 2017*; *Nielsen & Graves, 2017*; *United Nations Educational, 2018*).

This scarcity of data can be addressed by leveraging the vast amount of information generated through digital searches to examine internet user behavior, gain insights into specific aspects of community life, and ultimately support the development of public policies (*Aguirre et al., 2022*; *Rizzato et al., 2022*; *Di Profio et al., 2023*). In this domain, Google dominates structured search activity, serving as the primary tool for users seeking web content to meet their specific needs (*Bianchi, 2024*). A key advantage of such data lies in their quasi–real-time availability, which helps mitigate the delays commonly associated with traditional data collection, analysis, and forecasting processes (*Mavragani & Ochoa, 2019*). In this sense, data extracted from Google Trends hold considerable value for health research, particularly in monitoring public interest, supporting epidemiological outcomes, and enhancing the surveillance of symptoms and diseases (*Mavragani & Ochoa, 2019*; *Aguirre et al., 2022*; *Rizzato et al., 2022*; *Di Profio et al., 2023*). Within this context, the concept of *Infodemiology* has emerged, defined as "the science of distribution and determinants of information in an electronic medium, specifically the Internet, with the ultimate aim of informing public health and public policy" (*Eysenbach, 2009*).

Therefore, this study aimed to identify the interests of Google users in the terms "fake news," "misinformation," "disinformation," and "conspiracy theory," particularly in specific health-related contexts. We hypothesized that individuals from countries with higher average years of schooling (*H1*) and greater internet penetration (*H2*) would be more likely to search for these terms, as online information-seeking behavior is often associated with higher education levels and broader internet access (*Wang, Shi & Kong, 2020*).

## MATERIALS AND METHODS

The present manuscript incorporates excerpts from a previously published preprint (*Cruvinel et al., 2024*).

### Study design and ethical considerations

This longitudinal and retrospective ecological study collected computational metadata related to the interests of Google users from 25 countries regarding searches for the topics "fake news," "misinformation," "disinformation," and "conspiracy theory," as outlined in previous studies (*Mavragani & Ochoa, 2019*; *Aguirre et al., 2022*; *Rizzato et al., 2022*; *Di Profio et al., 2023*). Relative search volume (RSV) data were gathered from Google Trends for these four topics, covering all categories, for the period from January 2004 to March 2025. Data analyses were conducted according to the methods described below.

As this research utilized publicly available data, it was exempt from human subjects regulations and did not require institutional review board approval (*Lotto et al., 2023*).

## Relative search volume

Google Trends data can be collected anonymously and objectively, reducing biases commonly associated with inaccurate or misleading responses in structured interviews. Additionally, the platform allows information to be filtered by time and geographic regions, enabling rapid, cost-effective, and accurate analyses of internet users' information-seeking behavior. In this context, it is possible to extract metadata for specific topics, based on a Google algorithm that estimates the search volume of all relevant queries related to a given issue. This approach enables pattern comparisons across different countries.

Search trends for a given term are presented as a time series, showing weekly or monthly variations in the RSV index, scaled from 0 to 100. These values represent a normalized dataset, in which the maximum search interest within the selected period is assigned an index of 100 (SVI = 100), and all other values are calculated relative to this maximum.

On March 25, 2025, data were collected for the topics "fake news," "misinformation," "disinformation," and "conspiracy theory," filtered by Web search type across all categories, covering the period from January 2004 to March 2025.

## Mean years of schooling

The mean years of schooling for different countries were obtained from the Human Development Reports of the United Nations Development Program (*Human Development Reports, 2022*).

## Internet penetration

This variable was determined by the percentage of internet users in each country, as obtained from The World Bank database (*The World Bank, 2022*). Internet users are defined as individuals who have used the internet from any location *via* a computer, mobile phone, personal digital assistant, games machine, or digital TV in the last 3 months.

## Countries selection

The inclusion criteria consisted of countries with sufficient Google Trends data volume for the four topics related to information pollution, specifically those with RSV curves different from zero (RSV ≠ 0). Of the 250 countries available on Google Trends, 25 were selected for analysis: Australia (AUS), Brazil (BRA), Canada (CAN), Finland (FIN), France (FRA), Germany (DEU), Hong Kong (HKG), India (IND), Italy (ITA), Japan (JPN), Mexico (MEX), the Netherlands (NLD), Philippines (PHL), Poland (POL), Russia (RUS), Singapore (SGP), South Africa (ZAF), Spain (ESP), Sweden (SWE), Switzerland (CHE), Turkey (TUR), Taiwan (TWN), Ukraine (UKR), the United Kingdom (GBR), and the United States (USA).

## Qualitative categorization of queries

Two independent, trained, and calibrated investigators (OSJ and ML) analyzed queries based on their relationship to health issues. The training was conducted by an experienced researcher (TC), involving the assessment and discussion of queries. Subsequently, the investigators were calibrated through the evaluation of 100 queries, achieving an absolute inter-examiner agreement (ICC > 0.80) (*O'Connor & Joffe, 2020*). They manually coded the queries collected for the analysis regarding all categories as either non-health-related (code 0) or health-related (code 1). Codification as 1 was assigned only in cases of an unequivocal relationship of queries with health (*e.g.*, "vaccine" and "corona conspiracy theory"). Queries that were classified differently during the analysis were re-examined and discussed by the two investigators until a consensus was reached.

## Data analysis

Data were analyzed using Python 3 libraries within the Google Colaboratory environment. Seasonal decomposition was applied to identify the trend, seasonal, and residual components of the collected time series. In addition, Mann–Kendall tests were performed to assess the statistical significance of the trends identified through seasonal decomposition.

The frequencies of queries related to the four types of information pollution were weighted according to their RSV values. As the data did not follow a normal distribution, as determined by the Shapiro–Wilk test, the Spearman correlation test was applied to examine the relationships between the percentage of health-related queries and two variables: mean years of schooling and internet penetration.

The computational libraries used included *pandas, numpy, statsmodels, matplotlib. pyplot, pymannkendall, seaborn,* and *scipy.stats. P*-values < 0.05 were considered statistically significant.

## RESULTS

### Search volume trends

Figures 1–4 illustrate the time series curves representing Google users' interest in the topics "misinformation," "disinformation," "fake news," and "conspiracy theory" across different countries. The Mann–Kendall tests revealed distinct trends in user interest related to information pollution topics.

An increasing trend was observed for searches related to "fake news" in all countries. A similar pattern was found for "misinformation," except for France (FRA) and Japan (JPN), where no significant trend was detected. Interest in "disinformation" showed a decreasing trend in Italy (ITA) and an increasing trend in all other countries, except for France (FRA) and the United States (USA), where no trend was observed.

For the topic "conspiracy theory," decreasing trends were identified in Australia (AUS), Brazil (BRA), India (IND), Mexico (MEX), South Africa (ZAF), Spain (ESP), Turkey (TUR), and the United Kingdom (GBR), whereas increasing trends were observed in Canada (CAN), Finland (FIN), France (FRA), Germany (DEU), Hong Kong (HKG), Japan

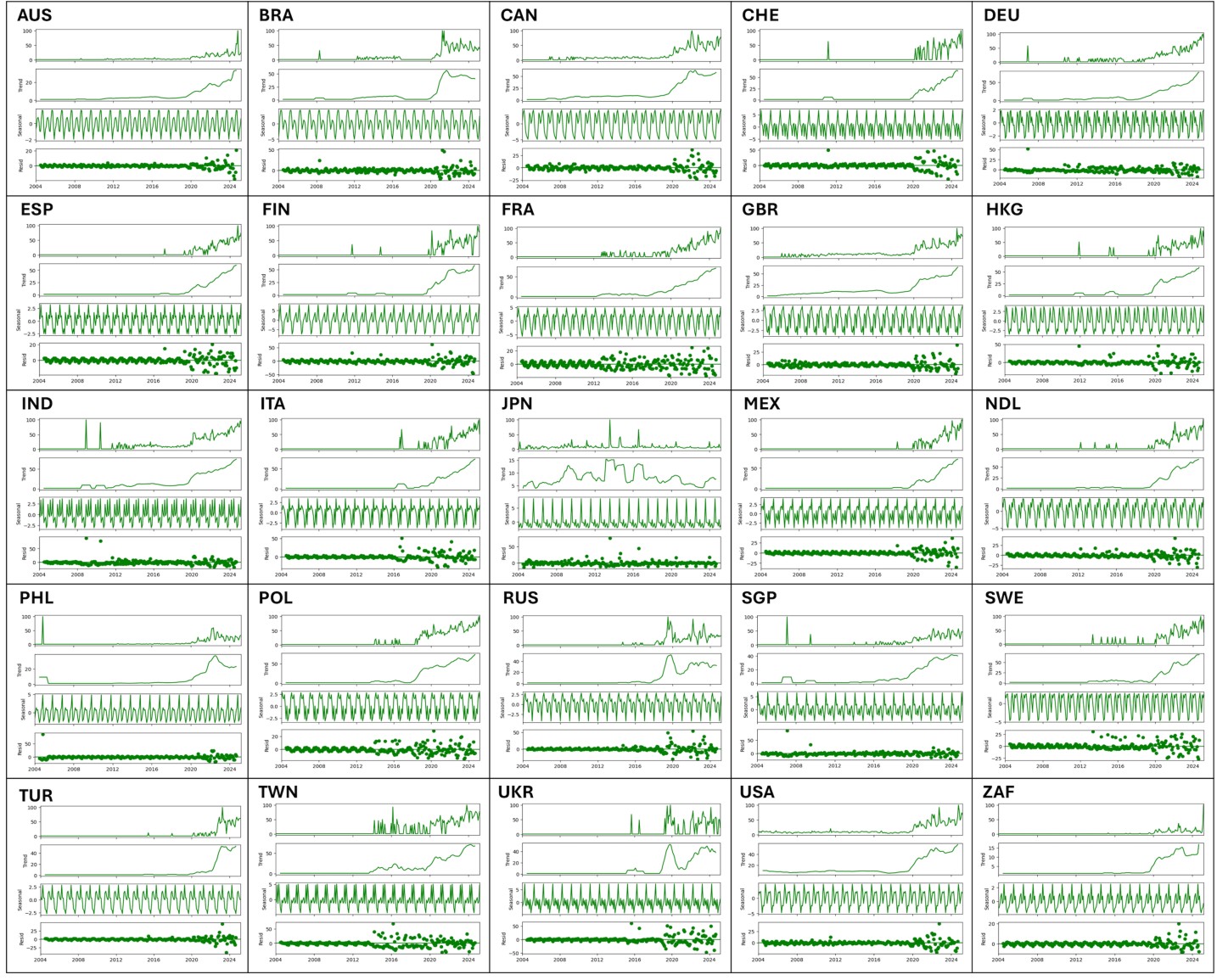

**Figure 1 Seasonal decomposition of time series for the topic of misinformation across countries.**

(JPN), the Netherlands (NLD), the Philippines (PHL), Poland (POL), Russia (RUS), Singapore (SGP), Sweden (SWE), Switzerland (CHE), Taiwan (TWN), Ukraine (UKR), the United Kingdom (GBR), and the United States (USA). No significant trend was observed in Italy (ITA) (Table 1).

## Internet penetration and education

Table 2 presents the internet penetration rates and mean years of schooling for each country, except for Taiwan (TWN), for which data were not available. The Netherlands (97%) and the United Kingdom (97%) had the highest internet penetration rates, while the Philippines (53%) and India (56%) reported the lowest. Regarding mean years of

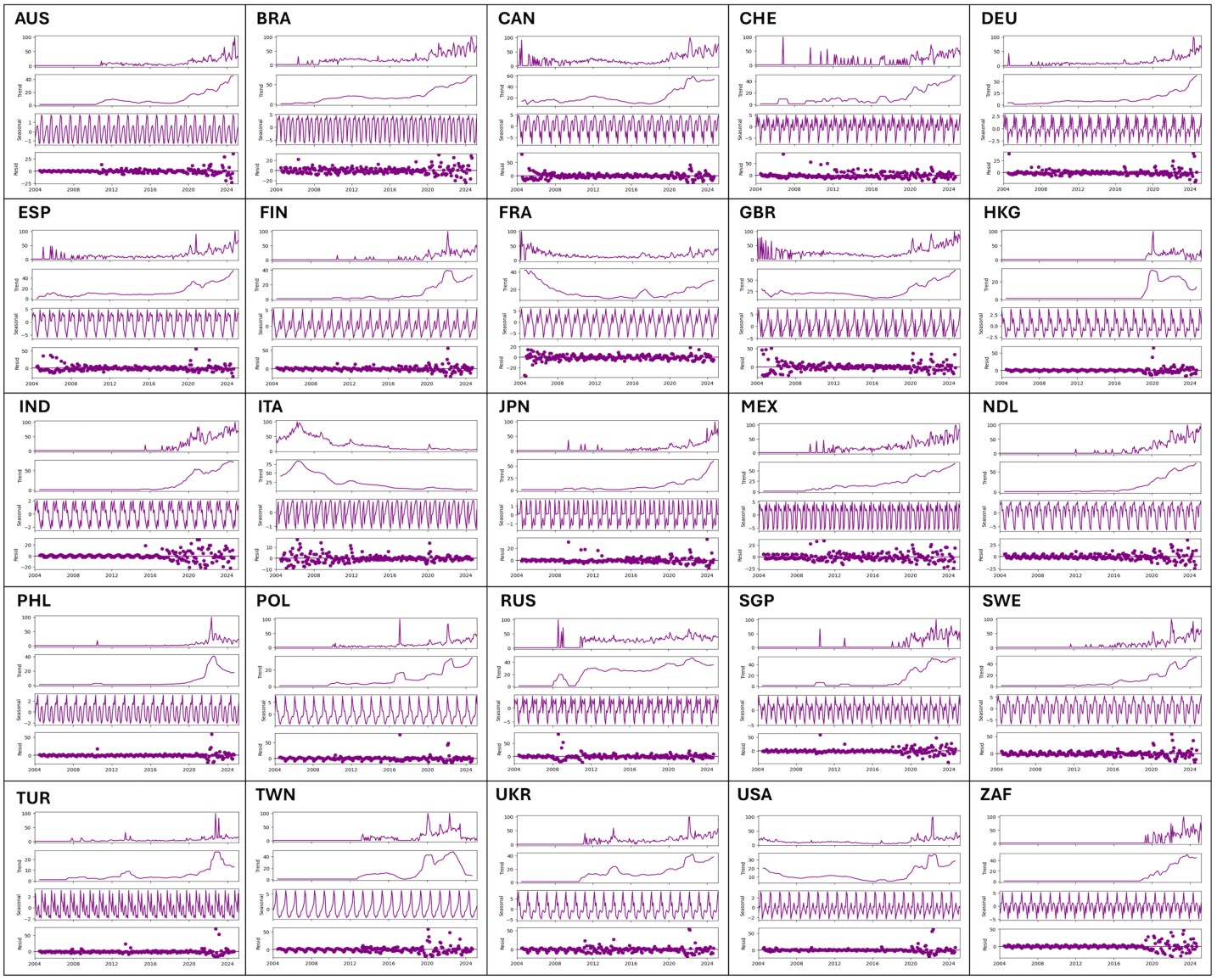

**Figure 2  Seasonal decomposition of time series for the topic of disinformation across countries.**

schooling, Germany (14.1 years) and Switzerland (13.9 years) had the highest averages, whereas India (6.6 years) and Brazil (8.1 years) had the lowest.

## Queries

In the analysis, 52 health-related queries were identified. Of these, seven were associated with the topic misinformation in 10 countries, five with disinformation in seven countries, 16 with fake news in 15 countries, and 24 with conspiracy theory in 17 countries (Table 3).

Regarding specifically health-related inquiries, all queries related to fake news were focused on COVID-19. Additionally, 71.4% of queries from the topic misinformation, 80% from the topic disinformation, and 79.1% from conspiracy theory were also linked to

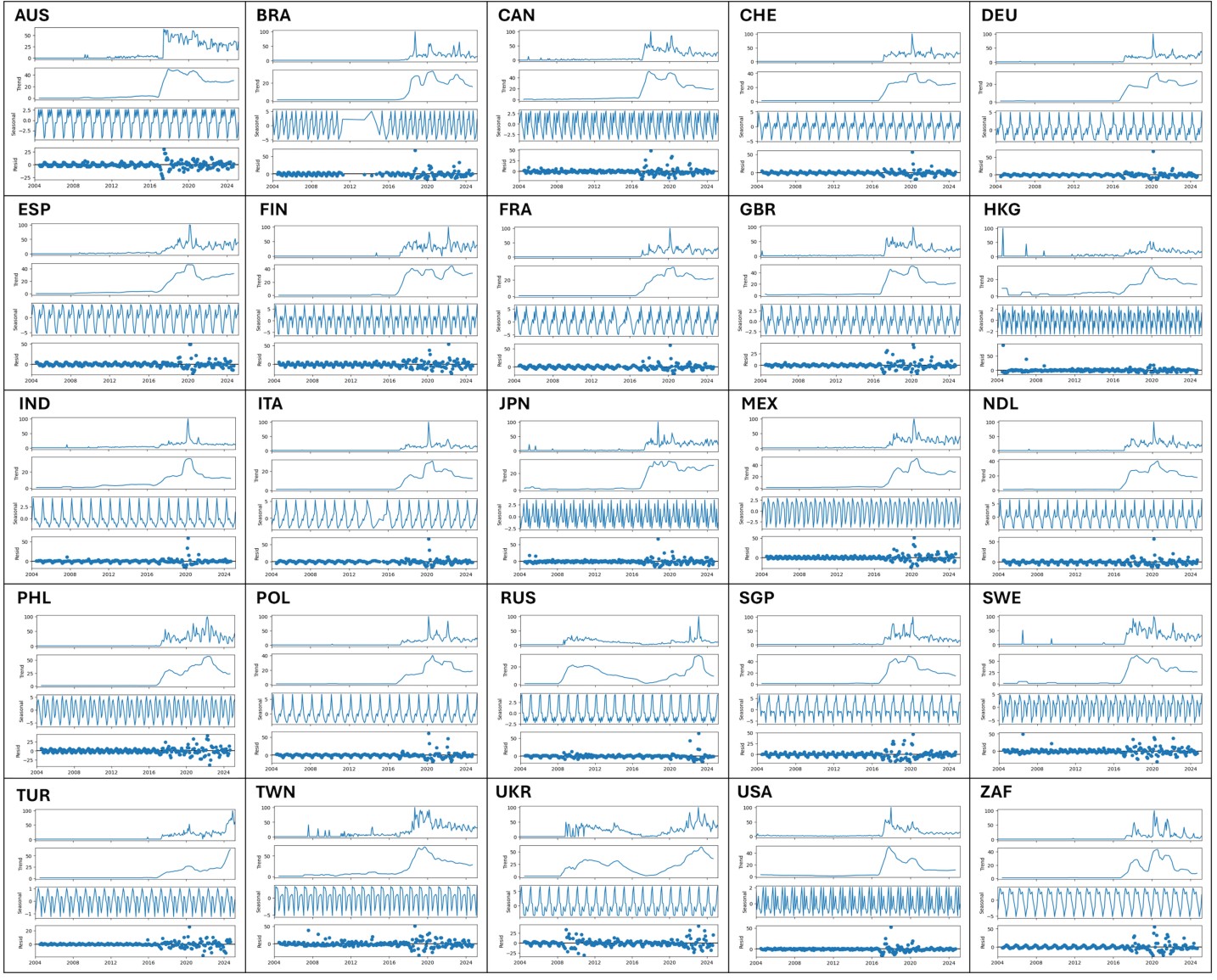

**Figure 3 Seasonal decomposition of time series for the topic of fake news across countries.**

COVID-19. The remaining queries were related to vaccines (14.2% for misinformation, 20% for disinformation, and 16.6% for conspiracy theory) and general health misinformation (14.2% for misinformation and 4.1% for conspiracy theory).

Figure 5 displays the correlations between the percentages of health-related queries and two variables: internet penetration and mean years of schooling. A significant positive correlation was observed between the percentage of health-related queries and internet penetration for misinformation ($r_s = 0.48$, $P = 0.017$). In contrast, correlations were not significant for disinformation ($r_s = -0.01$, $P = 0.952$), fake news ($r_s = 0.18$, $P = 0.408$), and conspiracy theory ($r_s = 0.20$, $P = 0.352$). Similarly, no significant correlations were found between the percentage of health-related queries and

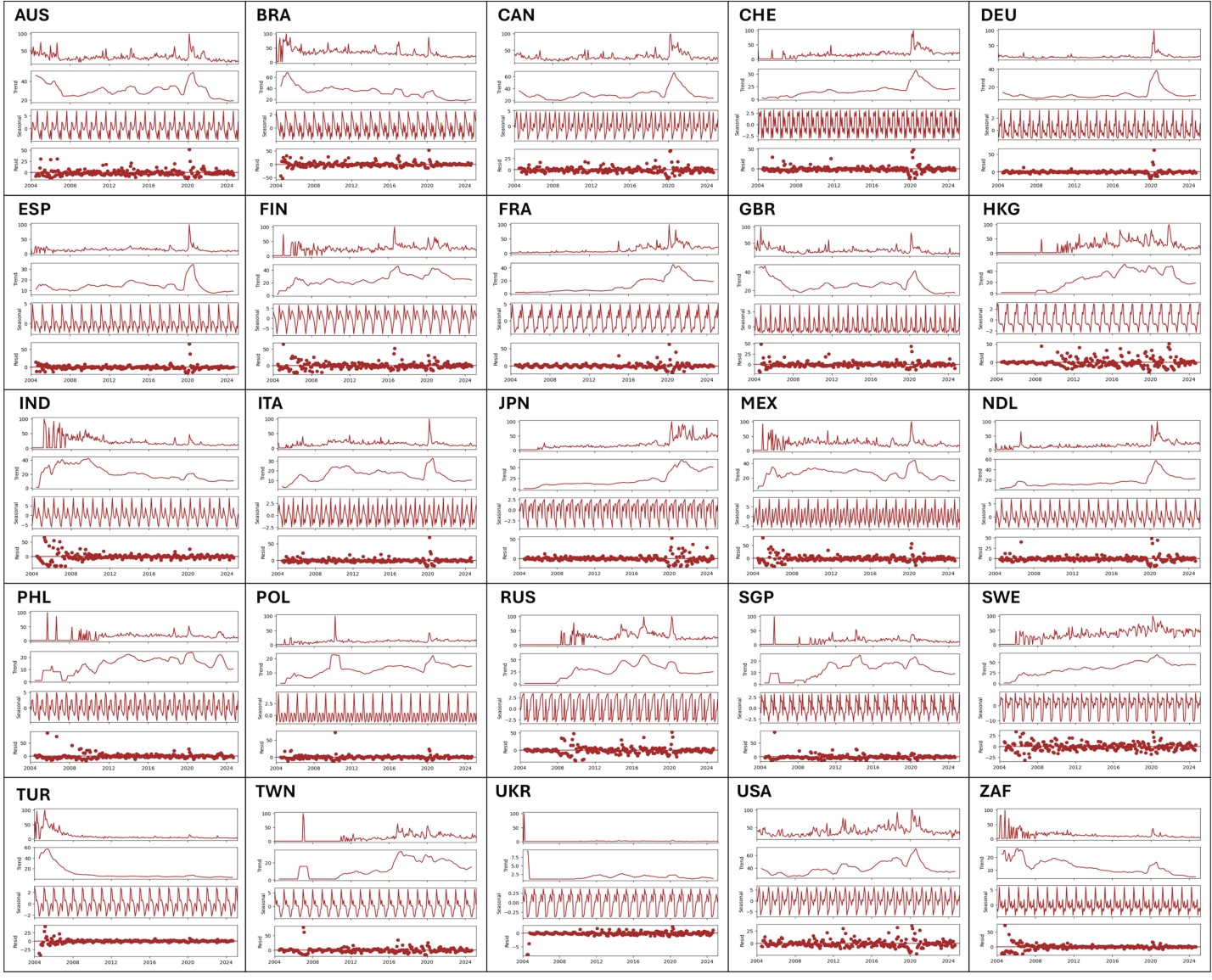

**Figure 4 Seasonal decomposition of time series for the topic of conspiracy theory across countries.**

mean years of schooling for misinformation (rs = 0.21, $P$ = 0.315), disinformation (rs = 0.33, $P$ = 0.121), fake news (rs = –0.02, $P$ = 0.927), and conspiracy theory (rs = 0.29, $P$ = 0.170).

## DISCUSSION

This study reveals that interest in terminologies related to information pollution has increased among Google users in most countries, particularly for the terms "fake news," "disinformation," and "misinformation" during the most recent years of the study period. Most health-related queries were associated with the COVID-19 pandemic, reinforcing the influence of global health crises on information-seeking behavior. Although interest in the

**Table 1 Trends in time series data identified by Mann–Kendall analysis.**

| Countries | Misinformation | | Disinformation | | Fake news | | Conspiracy theory | |
|---|---|---|---|---|---|---|---|---|
| | Z | P | Z | P | Z | P | Z | P |
| Australia | 16.56 | <0.001 | 13.82 | <0.001 | 13.13 | <0.001 | −4.37 | <0.001 |
| Brazil | 11.65 | <0.001 | 14.82 | <0.001 | 11.67 | <0.001 | −12.59 | <0.001 |
| Canada | 15.20 | <0.001 | 8.48 | <0.001 | 13.03 | <0.001 | 3.05 | <0.001 |
| Finland | 11.33 | <0.001 | 12.94 | <0.001 | 12.29 | <0.001 | 7.89 | <0.001 |
| France | 14.22 | 0.384 | −0.86 | 0.384 | 12.46 | <0.001 | 15.63 | <0.001 |
| Germany | 14.17 | <0.001 | 14.89 | <0.001 | 12.86 | <0.001 | 4.59 | <0.001 |
| Hong Kong | 11.93 | <0.001 | 10.72 | <0.001 | 13.22 | <0.001 | 9.60 | <0.001 |
| India | 15.21 | <0.001 | 13.98 | <0.001 | 15.13 | <0.001 | −8.64 | <0.001 |
| Italy | 12.09 | <0.001 | −17.63 | <0.001 | 11.87 | <0.001 | −0.91 | 0.360 |
| Japan | −0.29 | 0.766 | 14.60 | <0.001 | 14.10 | <0.001 | 16.37 | <0.001 |
| Mexico | 11.91 | <0.001 | 16.43 | <0.001 | 14.83 | <0.001 | −2.36 | <0.001 |
| Netherlands | 12.58 | <0.001 | 14.97 | <0.001 | 11.70 | <0.001 | 12.82 | <0.001 |
| Philippines | 16.29 | <0.001 | 13.50 | <0.001 | 12.91 | <0.001 | 5.35 | <0.001 |
| Poland | 14.33 | <0.001 | 16.52 | <0.001 | 13.20 | <0.001 | 10.16 | <0.001 |
| Russia | 13.10 | <0.001 | 13.71 | <0.001 | 4.63 | <0.001 | 6.00 | <0.001 |
| Singapore | 13.82 | <0.001 | 12.11 | <0.001 | 11.36 | <0.001 | 6.13 | <0.001 |
| South Africa | 12.40 | <0.001 | 10.99 | <0.001 | 11.52 | <0.001 | −9.06 | <0.001 |
| Spain | 12.44 | <0.001 | 13.33 | <0.001 | 16.63 | <0.001 | −2.83 | <0.001 |
| Sweden | 12.02 | <0.001 | 14.57 | <0.001 | 10.38 | <0.001 | 13.05 | <0.001 |
| Switzerland | 9.86 | <0.001 | 11.48 | <0.001 | 12.21 | <0.001 | 12.86 | <0.001 |
| Turkey | 10.89 | <0.001 | 13.94 | <0.001 | 14.19 | <0.001 | −13.57 | <0.001 |
| Ukraine | 9.10 | <0.001 | 14.90 | <0.001 | 8.99 | <0.001 | 7.92 | <0.001 |
| United Kingdom | 15.25 | <0.001 | 5.77 | <0.001 | 12.87 | <0.001 | −5.43 | <0.001 |
| United States | 9.59 | <0.001 | 1.67 | 0.093 | 8.03 | <0.001 | 5.69 | <0.001 |

**Table 2 Internet penetration (%) and mean years of schooling per country.**

| Countries | Internet penetration (%) | Schooling (mean years) |
|---|---|---|
| Australia | 96 | 12.7 |
| Brazil | 81 | 8.1 |
| Canada | 93 | 13.8 |
| Finland | 93 | 12.9 |
| France | 86 | 11.6 |
| Germany | 91 | 14.1 |
| Hong Kong | 96 | 12.3 |
| India | 56 | 6.6 |
| Italy | 75 | 10.7 |
| Japan | 83 | 13.4 |
| Mexico | 76 | 9.2 |
| The Netherlands | 97 | 12.6 |

| Countries | Internet penetration (%) | Schooling (mean years) |
|---|---|---|
| Philippines | 53 | 9.0 |
| Poland | 85 | 13.2 |
| Russia | 88 | 12.8 |
| Singapore | 94 | 11.9 |
| South Africa | 76 | 11.6 |
| Spain | 94 | 10.6 |
| Sweden | 88 | 12.6 |
| Switzerland | 96 | 13.9 |
| Turkey | 86 | 8.8 |
| Ukraine | 79 | 11.1 |
| United Kingdom | 97 | 13.4 |
| United States | 92 | 13.7 |

**Table 3  Health-related queries on information pollution topics by country.**

| Countries | Misinformation | Disinformation | Fake news | Conspiracy theories |
|---|---|---|---|---|
| AUS | COVID misinformation<br>Vaccine misinformation | – | Fake news COVID | Coronavirus conspiracy<br>COVID conspiracy theories<br>Conspiracy theories coronavirus |
| BRA | – | – | – | – |
| CAN | Misinformation COVID<br>Vaccine misinformation<br>Health misinformation<br>COVID-19 misinformation | COVID disinformation | Fake news COVID | COVID conspiracy theories<br>Coronavirus conspiracy |
| CHE | Misinformation related to the 2019–20 coronavirus outbreak | – | Fake news corona<br>Fake news coronavirus<br>COVID fake news | Corona<br>Coronavirus conspiracy theories<br>Coronavirus conspiracy theory |
| DEU | – | Disinformation corona | Corona fake news<br>Coronavirus fake news | Corona conspiracy theories<br>Conspiracy theory corona<br>Coronavirus conspiracy theories<br>Coronavirus conspiracy theory |
| ESP | – | – | – | Coronavirus<br>Coronavirus conspiracy |
| FIN | – | – | – | Corona conspiracy theory<br>Corona conspiracy theories |
| FRA | COVID misinformation | – | Fake news COVID<br>Coronavirus fake news | Coronavirus conspiracy<br>COVID conspiracy |
| GBR | Misinformation COVID<br>Misinformation COVID<br>Vaccine misinformation<br>Health misinformation | COVID disinformation | Coronavirus fake news<br>COVID fake news | Coronavirus conspiracy<br>COVID conspiracy theories |

| Countries | Misinformation | Disinformation | Fake news | Conspiracy theories |
|---|---|---|---|---|
| HKG | – | – | – | Vaccine |
| | | | | Vaccine conspiracy |
| IND | – | – | Corona fake news | – |
| ITA | – | Vaccines | Coronavirus fake news | Coronavirus conspiracy |
| | | Coronavirus disinformation | COVID fake news | COVID conspiracy |
| JPN | – | Corona misinformation | Corona fake news | Corona |
| | | | | Conspiracy corona |
| | | | | Vaccine |
| | | | | Vaccine conspiracy |
| | | | | Conspiracy theory corona |
| | | | | Vaccine conspiracy theory |
| | | | | Corona vaccine |
| | | | | Corona vaccine conspiracy |
| | | | | Corona virus conspiracy |
| | | | | Corona virus |
| MEX | Misinformation related to the 2019–20 coronavirus outbreak | – | – | – |
| NLD | COVID misinformation | – | Corona | Corona |
| | | | Corona fake | Corona conspiracy |
| | | | Corona fake news | Corona conspiracy theory |
| | | | Fake news corona | Corona conspiracy theory |
| | | | Corona news | |
| PHL | – | – | – | – |
| POL | – | – | Fake news coronavirus | Coronavirus conspiracy theories |
| | | | | COVID conspiracy theories |
| RUS | – | – | – | – |
| SGP | COVID misinformation | – | COVID fake news | COVID conspiracy theory |
| | Vaccine misinformation | | COVID-19 fake news | |
| | COVID-19 misinformation | | | |
| SWE | – | Disinformation corona | Corona | Conspiracy theories corona |
| | | | Corona fake | |
| | | | Corona fake news | |
| TUR | – | – | – | – |
| TWN | – | – | Pandemic fake news | Vaccine |
| | | | | Vaccine conspiracy |
| | | | | Vaccine conspiracy theory |
| | | | | Wuhan pneumonia |
| | | | | Wuhan pneumonia conspiracy |
| | | | | Pneumonia conspiracy theory |
| UKR | – | – | – | – |
| USA | COVID misinformation | COVID disinformation | – | – |
| | Vaccine misinformation | | | |
| | Health misinformation | | | |

| Countries | Misinformation | Disinformation | Fake news | Conspiracy theories |
| --- | --- | --- | --- | --- |
| ZAF | COVID-19 | – | COVID fake news | Coronavirus conspiracy |
| | | | COVID-19 | COVID conspiracy theories |
| | | | Fake news COVID 19 | Coronavirus conspiracy theories |
| | | | COVID-19 | |
| | | | Fake news about COVID | |
| | | | Fake news about COVID 19 | |
| | | | Coronavirus fake news | |
| | | | Corona fake news | |
| | | | Fake news about COVID-19 | |

term "conspiracy theory" also increased in many countries, a significant number demonstrated a decreasing trend over time, possibly reflecting saturation or shifting public engagement with the term. Based on these findings, hypothesis *H1* was rejected, and hypothesis *H2* was partially accepted.

As previously demonstrated, internet penetration is positively associated with health information-seeking behavior (*Wang, Shi & Kong, 2020*), which may explain the moderate correlation observed between internet penetration and the volume of health-related queries specifically for "misinformation." The broader diversity of health-related queries within this category—compared to the others—may account for why this correlation was only significant in this case.

While the spread of falsehoods is not a new phenomenon, discourse around information pollution gained particular prominence following the 2016 U.S. presidential election and the COVID-19 pandemic (*Bovet & Makse, 2019*; *Tangcharoensathien et al., 2020*). These historical moments likely amplified public exposure to terms such as "fake news" and "disinformation," which became more frequent in mass media and on the web. Social media platforms often serve as the first point of contact for such terms, with user engagement shaped by individual factors like pre-existing beliefs, limited critical thinking skills, and ideological alignment (*Vosoughi, Roy & Aral, 2018*; *Giglietto et al., 2019*; *Wang et al., 2019*; *Scheler & Pennyccok, 2020*). Additionally, the structural characteristics of these platforms—such as algorithmic prioritization and the monetization of high-engagement content—exacerbate the spread of misleading or false information (*Rodrigues et al., 2024*).

This environment can lead individuals to misuse information pollution terminology to discredit narratives that challenge their beliefs, as seen in the politically motivated rejection of scientific discourse surrounding COVID-19 vaccines (*Buchanan, 2020*). Among the topics studied, conspiracy theories appear to exert a unique influence, engaging users across political and health-related domains, and even in entertainment contexts (*Jolley, Marques & Cookson, 2022*). This broader cultural embedding may explain both the early widespread use of the term and the declining interest observed in several countries—users may have exhausted their interest in the topic *via* traditional search engines, having engaged with it through other platforms or contexts such as books and films.

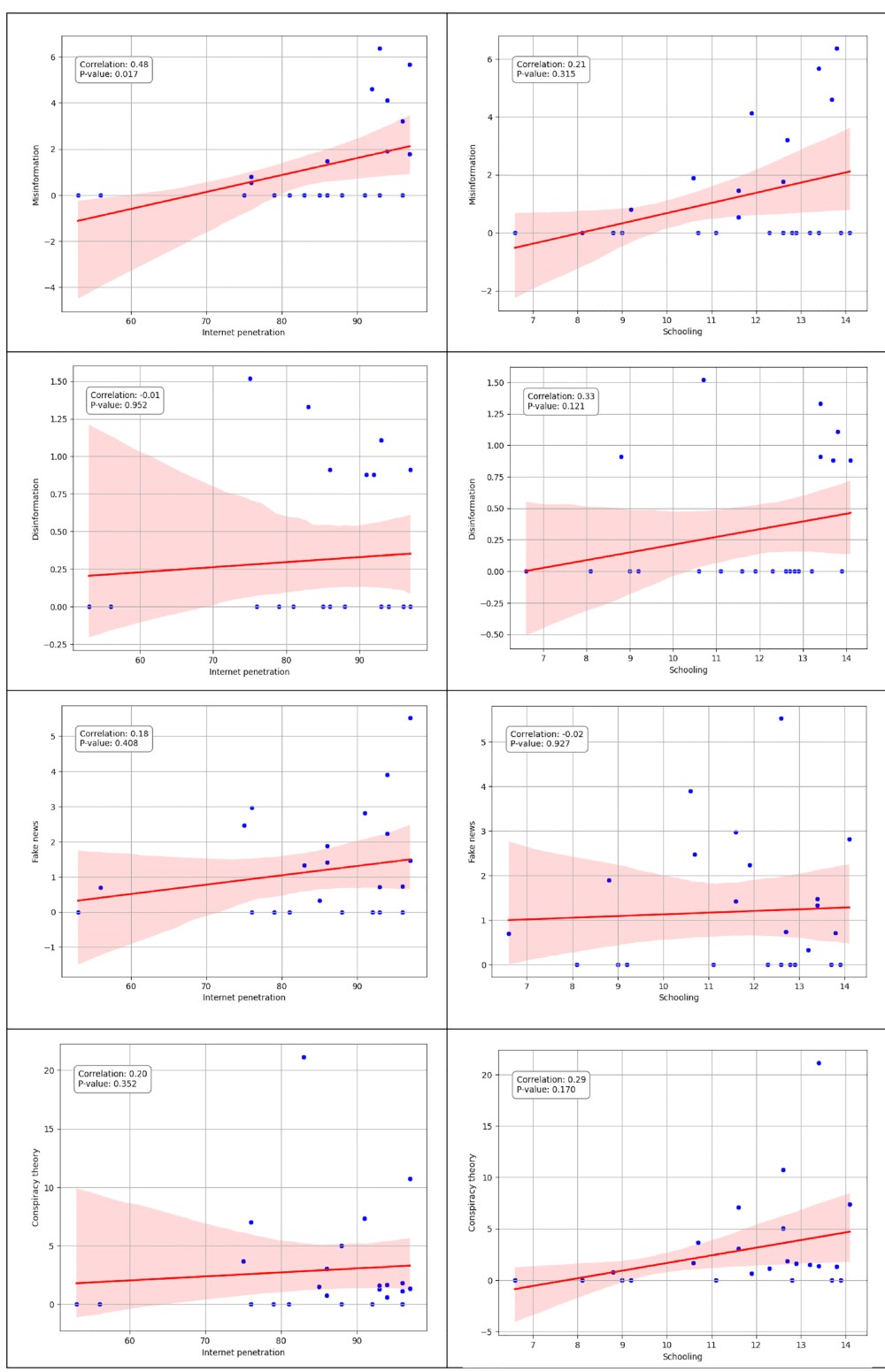

**Figure 5  Scatterplots showing the correlation between the relative search volumes of health-related queries and internet penetration and mean years of schooling.**

Importantly, this study employed Google-provided topics to ensure comparability across countries, reducing the risk of bias related to search term selection. Although the opaque nature of Google's algorithms introduces some limitations, the qualitative analysis of queries confirmed alignment with the study's aims. The prominence of health-related queries during the COVID-19 pandemic reinforces concerns about the widespread circulation of disinformation during global crises and highlights the public's demand for clarifying information during such periods.

Some methodological considerations should be acknowledged. First, inclusion in the analysis depended on the availability of Google Trends data, which excluded some countries due to insufficient data volume. Notably, most of the analyzed countries had high internet penetration, with only a few (India, Italy, Mexico, the Philippines, South Africa, and Ukraine) reporting values below 80%, and a median penetration rate of 88%. Additionally, search behaviors may be affected by national policies restricting internet freedom, such as in China and Russia (*Freedom House, 2023*), potentially explaining the absence or inconsistency of data from these regions. Finally, health-related queries were only classified as such if they were directly linked to health issues, which may have resulted in an underestimation of relevant searches in the initial phase of data coding.

## CONCLUSIONS

Overall, this study supports the increasing interest of Google users in terms related to information pollution over time, although the association of these terms with health-related queries appears limited. This underscores the importance of implementing media education—particularly in schools—as a long-term strategy to promote a better understanding of key concepts among the population and to encourage the pursuit of information from diverse and reliable sources. Moreover, international organizations should actively monitor and regulate the emergence and spread of new terminology related to information pollution, especially in the context of growing internet penetration worldwide.

## ACKNOWLEDGEMENTS

We used ChatGPT to review the formatting and grammar of our manuscript.

### Funding
This work was supported by the São Paulo Research Foundation (Grant #2023/02547-9). The funders had no role in study design, data collection and analysis, decision to publish, or preparation of the manuscript.

### Grant Disclosures
The following grant information was disclosed by the authors:
São Paulo Research Foundation: 2023/02547-9.

## Competing Interests

The authors declare that they have no competing interests.

## Author Contributions

- Thiago Cruvinel conceived and designed the experiments, performed the experiments, analyzed the data, prepared figures and/or tables, authored or reviewed drafts of the article, and approved the final draft.
- Olívia Santana Jorge conceived and designed the experiments, analyzed the data, prepared figures and/or tables, authored or reviewed drafts of the article, and approved the final draft.
- Matheus Lotto conceived and designed the experiments, analyzed the data, performed the computation work, prepared figures and/or tables, authored or reviewed drafts of the article, and approved the final draft.
- Bruna Nogueira performed the experiments, prepared figures and/or tables, authored or reviewed drafts of the article, and approved the final draft.
- Ana Maria Jucá performed the experiments, prepared figures and/or tables, authored or reviewed drafts of the article, and approved the final draft.
- Agnes Cruvinel conceived and designed the experiments, prepared figures and/or tables, authored or reviewed drafts of the article, and approved the final draft.

## Data Availability

The data was collected from Google Trends (https://trends.google.com.br/trends) is available at Figshare: Lotto, Matheus; Santana Jorge, Olívia; Cruvinel, Thiago (2025). Raw data of the manuscript "Interests of Google users in health-related issues concerning information pollution terminologies: An infodemiology study". figshare. Dataset. https://doi.org/10.6084/m9.figshare.28850879.v1.

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
