# Peer review of "Interests of Google users in information pollution terminologies: an infodemiology study"

_PeerJ Computer Science, doi:10.7717/peerj-cs.2912_

## Round 0.1 · original submission · Major Revisions

We have received three reviews that highlight strengths and weaknesses of the manuscript. Although the paper has significant potential, it requires major revisions to meet the required standards.
I summarize the main issues to be addressed:
Definitions and classifications: It is necessary to provide clearer and more rigorous definitions of the categories of "disinformation", "fake news", "misinformation" and "conspiracy theories". The categories seem to overlap, as highlighted in the third review, compromising the robustness of the analysis. It is also suggested to include an additional category, such as "Neutral Information", to improve the classification.

Analysis and interpretation of results: In several sections (e.g. "Search Volume Trends" and "Queries"), the results are not sufficiently analyzed. Provide a clear and detailed interpretation of the observations reported in the figures and data, and include a more in-depth analysis of the main queries identified.
Methodology and transparency: Although the use of ARIMA models and chi-square tests is methodologically sound, more clarity is needed on the data used. For example, explain how the Google Trends data was interpreted (absolute or relative values) and which queries were included in the analyses. Also, highlight any overlap between categories.

Missing data: Provide the raw data, as requested in the third review, to ensure transparency and allow reviewers to fully assess the robustness of the results.

Context and literature: Improve the connection to existing literature, especially with respect to previous studies that use Google Trends data for time-series research. This would strengthen the relevance and position of your work within the field.

Errors and omissions: Correct reported inaccuracies, such as the exclusion of significant countries (India and China) and the incorrect citation of Internet penetration rates. These errors reduce the overall credibility of the study.

Reviewer 1 ·

Basic reporting

1. Overall, the language use is clear and professional, but a few grammatical and structural problems need to be fixed (such as line 84: “accordingly previous studies”).
2. The literature did not provide sufficient references. It did not clearly define information disorders. Also, the categorization of misinformation, disinformation, conspiracy theories, and fake news is debatable. The authors seek to compare search trends across four types of information disorders; however, the distinctions between these categories are not always clear or mutually exclusive. For instance, consider the claim, “COVID-19 vaccines contain microchips to track people.” This example could fall under both misinformation and fake news, as false information may spread through well-meaning individuals (misinformation) or deliberately fabricated news (fake news). Similarly, the statement, “NASA faked the moon landing as part of a government conspiracy,” underscores how conspiracy theories often involve elements of disinformation. Since these categories frequently overlap in practice, despite their distinct theoretical meanings, the study's contribution becomes ambiguous. The lack of clarity around how these categories are treated raises questions about the robustness of the analysis.
3. Raw data was not provided.

Experimental design

1. Methods: In lines 99 to 105, the authors describe using Google Trends to track topics rather than keywords, explaining that topics are algorithmically determined by Google. While this is a valid approach, it is unclear what specific queries are encompassed within each topic. Are these queries consistent across different countries, or do they vary? If so, what are these variations, and is there overlap between topics like misinformation and fake news? For example, could a query be categorized under both “misinformation” and “fake news”? If this overlap exists, it undermines the analytical distinction between these categories, potentially making the findings less reliable.
2. Analysis: The authors use chi-square tests to examine whether countries with higher schooling years or greater internet penetration tend to search more for specific information disorder terminologies. However, the analysis appears misaligned with the stated hypothesis. The hypothesis suggests that higher levels of schooling should predict increased search interest in each type of information disorder, but the chi-square test only assesses the relationship between schooling (or internet penetration rate) and the type of disorder searched for, not the volume or likelihood of searches.

Validity of the findings

1. Considering the wide disparities among the 21 countries included in the analysis—such as differences in economic, political, and cultural contexts—relying solely on schooling years or internet penetration as predictors without accounting for these other factors limits the validity and depth of the findings.
2. Underlying data was not provided.
3. Due to inappropriate analysis, the results did not answer the original research hypothesis.

Additional comments

For further detailed comments, please refer to the annotated manuscript.

Annotated reviews are not available for download in order to protect the identity of reviewers who chose to remain anonymous.

Reviewer 2 ·

Basic reporting

This is a compelling article. The writing is unclear in places, though, and the authors should better contextualize this paper relative to communication research.

With regard to writing, sentences such as the following are unclear: "Using Google Trends, relative search volumes (RSVs) that disclosed topics such as 'disinformation,' 'fake news,' 'misinformation,' and 'conspiracy theory' were collected in all and health categories, from January 2004 to March
2023." What does "and health categories" mean? Also, avoid passive voice. Instead say something like we assessed the relative search volumes of keywords such as XXXX from January 2004 to March 2025.

In terms of contextualizing the paper, the authors should consider connecting this paper to literature which has used Google trends data in various ways in time-series research.

Beyond that, part of what is compelling about this paper is the way in which the authors think about how people think about misinformation as a concept worth investigating. In light of that, the authors also might want to connect this paper more directly to recent literature on how we should think about and define misinformation.

Experimental design

The authors employ time-series design here in an interesting way. ARIMA modeling is an appropriate approach.

Validity of the findings

The results the authors present are straightforward and understandable. The descriptive results suggest low prevalence of searching. As for prediction of that searching, there is one concern the authors should address. I have understood Google trend data to be relative in that it shows the relative rise or fall of search compared to a baseline unique to the defined time window in the defined geographical context. I did not think absolute values of searches were available, which suggests that country-to-country comparison of absolute search levels would not be possible and yet the authors talk about comparing countries in terms of, for example, "greater frequency of searches." Are the numbers in Table 2 actually total searches? If so, please clarify how the research obtained that data. If not, please consider what exactly is being predicted as dependent variable. That needs to be clarified.

Additional comments

See comments above.

·

Basic reporting

Formal results should include clear definitions of all terms and theorems, and detailed proofs.: Missing.

In the section "Results → Search Volume Trends", the observations depicted in Figures 1 and 2 are left unexplained.
Providing a clear analysis of what is observed in these figures would greatly enhance the clarity and interpretability of this section.

In the section "Results → Queries", it is mentioned that "84 health-related queries were identified, of which 46 were associated with 193 conspiracy theories."
However, the specific queries are not listed. Highlighting the major queries would improve the transparency and depth of the analysis.

Experimental design

Strengths:

1. Including data from 21 countries with diverse internet penetration rates and educational levels offers a comprehensive and broad perspective on the phenomenon being studied.

2. The study effectively addresses the critical issue of information disorders, an area of growing importance, especially in light of the COVID-19 pandemic and its associated misinformation challenges.

3. The application of ARIMA forecasting models for predictive analysis, alongside chi-square tests for statistical significance, demonstrates a robust and well-structured methodological approach.

Weaknesses:

1. The exclusion of India, the world's most populous country with an internet penetration rate of 52.4%, raises significant questions.
Additionally, the results section inaccurately lists India's internet penetration rate as 46%, necessitating correction.
Furthermore, the source of data in Table 1 is unclear.
Also China's internet penetration rate is cited as 78%, yet both India and China, the two most populous countries, are omitted.
Their inclusion could potentially alter the research findings significantly.

2. In the section "Results → Search Volume Trends", the observations depicted in Figures 1 and 2 are left unexplained.
Providing a clear analysis of what is observed in these figures would greatly enhance the clarity and interpretability of this section.

3. In the section "Results → Queries", it is mentioned that "84 health-related queries were identified, of which 46 were associated with 193 conspiracy theories."
However, the specific queries are not listed. Highlighting the major queries would improve the transparency and depth of the analysis.

4. In the "Introduction", four types of information disorders are listed: misinformation, fake news, disinformation, and conspiracy theories.
However, a fifth type, "Neutral Information", is overlooked.
Neutral Information refers to types of information that cannot be verified with current methods or data.
For instance, the claim that "COVID-19 originated from Wuhan Labs" represents a case of Neutral Information.
Including this category would provide a more complete classification.

Overall Suggestion:
Resubmit after addressing the weaknesses highlighted in this review.

Validity of the findings

As above

Additional comments

As above

Cite this review as

---

## Round 0.2 · accepted · Accept

The manuscript is now suitable for publication. The authors have satisfactorily addressed all the concerns raised in the previous review, and the revisions have significantly improved the clarity and completeness of the work. I recommend acceptance in its current form.

Reviewer 2 ·

Basic reporting

The authors have improved the clarity of their manuscript.

Experimental design

The authors use a time-series design, which is appropriate here.

Validity of the findings

The authors have revised their analysis and now more appropriately report results.

Additional comments

The authors include some errors, e.g., Pennyccok, and so should conduct one more round of copy editing. They also have expanded the literature cited though could additionally consider time-series work on Google searches such as Weeks, B. E., Friedenberg, L. M., Southwell, B. G., & Slater, J. S. (2012). Behavioral consequences of conflict-oriented health news coverage: The 2009 mammography guideline controversy and online information seeking. Health Communication, 27(2), 158-166.